# The AMPK/p27^Kip1^ Pathway as a Novel Target to Promote Autophagy and Resilience in Aged Cells

**DOI:** 10.3390/cells10061430

**Published:** 2021-06-08

**Authors:** Lauren K. McKay, James P. White

**Affiliations:** 1Adams School of Dentistry, UNC Chapel Hill, Chapel Hill, NC 27599, USA; lauren_katz@unc.edu; 2Duke Molecular Physiology Institute, Duke University School of Medicine, 300 N. Duke Street, Durham, NC 27701, USA; 3Department of Medicine, Duke University School of Medicine, 300 N. Duke Street, Durham, NC 27701, USA; 4Duke Center for the Study of Aging and Human Development, Duke University School of Medicine, 300 N. Duke Street, Durham, NC 27701, USA

**Keywords:** p27, AMPK, autophagy, apoptosis, aging, Akt, senescence

## Abstract

Once believed to solely function as a cyclin-dependent kinase inhibitor, p27^Kip1^ is now emerging as a critical mediator of autophagy, cytoskeletal dynamics, cell migration and apoptosis. During periods of metabolic stress, the subcellular location of p27^Kip1^ largely dictates its function. Cytoplasmic p27^Kip1^ has been found to be promote cellular resilience through autophagy and anti-apoptotic mechanisms. Nuclear p27^Kip1^, however, inhibits cell cycle progression and makes the cell susceptible to quiescence, apoptosis, and/or senescence. Cellular location of p27^Kip1^ is regulated, in part, by phosphorylation by various kinases, including Akt and AMPK. Aging promotes nuclear localization of p27^Kip1^ and a predisposition to senescence or apoptosis. Here, we will review the role of p27^Kip1^ in healthy and aging cells with a particular emphasis on the interplay between autophagy and apoptosis.

## 1. Introduction

The kinase inhibitory protein p27 (p27^Kip1^) is widely known for its canonical role as a cyclin-dependent kinase (CDK) inhibitor. Specifically, p27^Kip1^ binds to and inhibits the protein cyclin E/CDK2, resulting in G0/G1 cell cycle arrest, ultimately blocking cell proliferation and inducing cell quiescence [1,2,3]. Due to its ability to promote cell cycle arrest and inhibit proliferation, p27^Kip1^ has been extensively studied for its role in cancer as a tumor suppressor; however, several novel functions of p27^Kip1^ have been recently discovered that are independent of its role in CDK inhibition and cell cycle control. These functions include mediating cytoskeletal dynamics, cell migration, and apoptosis and autophagy [4,5]. Such diverse roles of p27^Kip1^ are regulated by transcription [6], phosphorylation [5,7,8], degradation [7,8,9], and subcellular location [5,7,8]. As an intrinsically disordered protein (IDP), various phosphorylations most frequently dictate the specific function of p27^Kip1^. Sites of modification include threonine 187 (Thr187), serine 10 (Ser10), and threonine 198 (Thr198)s. The phosphorylation of Thr187 by CDK2 results in cell cycle progression through degradation of p27^Kip1^ inside the nucleus [10,11], while Ser10 phosphorylation by hKIS promotes transport of p27^Kip1^ out of the nucleus, allowing for cell cycle progression [12]. Additionally, Ser10 phosphorylation by CDK5 contributes to actin organization and cortical neuronal migration via cytoplasmic stabilization of p27 [4]. Akt will phosphorylate p27^Kip1^ at Thr157 to promote cytoplasmic localization, stability, and cell cycle progression [7,13,14]. Most pertinent to this review is 5′ AMP-activated protein kinase (AMPK)-dependent phosphorylation of p27^Kip1^ on Thr198, which promotes sequestration of p27^Kip1^ to the cytosol, resulting in a reduction of apoptosis and activation of autophagy [5]. A summary of upstream p27^Kip1^ signaling is illustrated in Figure 1. Based on the aforementioned functions, it is evident that subcellular location of p27^Kip1^ is integral to its function. In the context of apoptosis and autophagy, it has been observed that nuclear p27^Kip1^ promotes quiescence, apoptosis, and senescence, while cytoplasmic p27^Kip1^ enhances cell survival and autophagy [15]. Furthermore, there is evidence that aging affects the dynamic between autophagy and apoptosis, with a reduction in autophagy and an increase in cell death in older cells as a result of greater expression of nuclear p27^Kip1^ [13,15,16]. Here, we will review the diverse functions of p27^Kip1^ in the context of autophagy and apoptosis with a focus on age-related changes in p27^Kip1^ expression, location, and function.

## 2. Apoptosis and Autophagy

Activation of the autophagy pathway was first discovered under nutrient deprivation to provide substrates for protein synthesis and TCA intermediates during periods of negative nutritional stress [14]. Such substrates are generated through degradation of organelles and proteins. Damaged organelles and misfolded proteins are engulfed by autophagosomes and delivered to lysosomes for degradation. The released products are then recycled as an alternate energy source and substrates for protein synthesis. While autophagy is an ongoing cellular process, we now know that it is significantly upregulated during various cell stresses including exercise, ER stress, infection, hypoxia, and oxidative damage [17,18].

If cells are unable to recover from the respective stress, apoptosis, or programmed cell death is a common fate. Multicellular organisms undergo the physiological process of apoptosis as a way to eliminate damaged cells and maintain tissue integrity. Apoptosis can result from cell cycle arrest in the late G1 or S phase [19,20,21]. p27^Kip1^ is known to function by blocking the transition from G1 to S phase and therefore plays an important role in apoptosis. When p27^Kip1^ was overexpressed in lung cancer cell lines, apoptosis was induced through downregulation of pRb expression [22]. Additionally, spontaneous apoptosis was significantly higher in p27-positive tumors from individuals with oral and oropharyngeal squamous cell carcinoma compared to p27-negative tumors. p27-positive tumors were also associated with higher levels of Bax expression, an apoptosis-related protein [23].

While p27^Kip1^ is considered to be pro-apoptotic in carcinogenesis [24], there is evidence that p27^Kip1^ protects cells from apoptosis during conditions of cellular stress. Apoptosis can be induced via the activation of CDKs [20,25,26,27,28,29]. As a Cdk inhibitor, p27^Kip1^ has the ability to prevent apoptosis by directly regulating CDK-2 activation [30,31]. Using p27^Kip1^-deficient cells, Hiromura et al. demonstrated that both CDK-2 activity and rates of apoptosis were elevated during serum deprivation-induced cell stress. Apoptosis was prevented by either restoring p27^Kip1^ expression or decreasing the activity of CDK-2 [31]. In connection with protecting cells from apoptosis, p27^Kip1^ has also been shown to promote autophagy during periods of metabolic stress.

The interplay between apoptosis and autophagy can be seen across many cell types [32]. The relationship between cell death and survival is vital to maintaining cellular homeostasis. Disruption of this balance results in pathophysiological consequences including cancer and autoimmune disease. Furthermore, it has been found that there are age-associated alterations affecting the interchange between these two pathways, which will be discussed in detail later in the review. In response to stress, a cell will adapt to the stress (autophagy) or undergo programmed cell death (apoptosis). Generally, autophagy precedes apoptosis [33]. A cell will initially activate autophagy in an attempt to survive stress, but will undergo apoptosis if this mechanism fails. These two processes cross-regulate each other, in that the activation of autophagy blocks the apoptotic program, while apoptosis suppresses the induction of autophagy [34]. The interplay between these two mechanisms primarily occurs through Bcl-2, an inhibitor of cell death. Cell survival is promoted when Bcl-2 interacts with the PI3K complex of the autophagy pathway. Additionally, Bcl-2 prevents apoptosis by inhibiting the pro-apoptotic member Bax; however, when the JNK pathway is activated, Bcl-2 will become phosphorylated and lose its ability to bind Bax, resulting in apoptosis [32].

During times of prolonged nutrient deprivation, cell fate is determined by the interplay between apoptosis and autophagy. Despite being characterized as a cyclin inhibitor, p27^Kip1^ is now known to play a critical role in the mediation of cell fate during metabolic stress conditions [5]. The induction of autophagy or apoptosis programs are two common responses to cellular stress. As p27^Kip1^ can be involved in both processes, this review will dissect mechanism pertaining to each process.

## 3. AMPK and Cellular Energy Homeostasis

AMPK is a serine/threonine kinase existing as a heterotrimer. It is composed of a 63 kDa catalytic (α) subunit, and two non-catalytic regulatory subunits, β (30 kDa) and γ (38–63 kDa), in a ratio of 1α:1β:1γ [35,36]. The stability and function of the AMPK complex requires all three subunits [37,38]. Each subunit has slight alterations in structure to allow for different functions to regulate AMPK activity. The α subunit is composed of a highly conserved N-terminal catalytic domain containing the phosphorylation site at Thr172, an auto inhibitory domain [39,40] and a C-terminus containing binding sites for the β and γ subunits [39,41]. AMPKα1 is ubiquitously expressed through tissues while AMPKα2 is dominant in skeletal muscle, heart, and liver [42,43]. Although still indecisive, the α1 and α2 subunits appear to have different functions and localization. AMPKα1 is primarily localized in the cytoplasm, while AMPKα2 is localized in the cytoplasm and nucleus [44]. Furthermore, it has been theorized that the AMPKα2 subunit may be involved in regulating gene expression. In human skeletal muscle, 60 min of exercise resulted in an increase in nuclear translocation of AMPKα2 [45]. These results support the notion of AMPK α subunits regulating cell signaling along with gene expression. The β subunit serves as scaffolding for the AMPK complex containing an N-terminal myristoylation site that can target AMPK to transmembranes [46,47], a glycogen binding domain [48], and a C-terminal binding domain for α and γ subunits. The α and γ bind domains appear to be sufficient to stabilize the AMPK heterotrimers [37]. The β subunit also contains phosphorylation sites for the regulation of catalytic activity (Ser182) and nuclear localization (Ser24 and Ser25) [46]. The intracellular localization of the different AMPK subunits has suggested that AMPK can regulate gene expression and/or cytosolic signaling.

AMPK activation is regulated by several mediators. AMP can allosterically activate AMPK or by phosphorylation by one or more upstream kinases at a threonine residue within the activation loop of the α subunit [49,50]. The phosphorylation state of AMPK is increased by a combination of AMP-induced increases in upstream kinases and a reduction of phosphatases [51,52]. The multiple targets that AMP can activate will induce a large activation in the activity of AMPK with relatively small changes in AMP. The energy state of the cell is not solely monitored by AMP concentrations. High ATP concentrations will serve as an antagonist of processes activated by high AMP levels. Thus, the AMP:ATP ratio appears to the critical regulator of AMPK activity.

AMPK plays a key role in cellular energy homeostasis. AMPK is important to promote energy availability while ensuring cell survival [53]. AMPK has numerous downstream targets [54], several of which regulate autophagy and apoptosis. In the setting of energy deficiency, AMPK can promote autophagy directly through phosphorylation of ULK1 to initiate the formation of autophagosomes [55]. AMPK also has the ability to regulate autophagy indirectly by inhibition of mTORC1 through phosphorylation of the tuberous sclerosis complex 2 (Tsc2) [56,57,58,59,60] and/or through phosphorylation of raptor [61].

## 4. AMPK/p27^Kip1^ Signaling Promotes Cell Survival and Autophagy

Mediation of cell fate during states of metabolic stress is an established function of p27^Kip1^. Of particular interest is that the subcellular location of p27^Kip1^ largely dictates its function in the apoptosis and autophagy pathways [5,15,62,63,64]. The cellular location of p27^Kip1^ has been described as dynamic, localizing between the nucleus and the cytoplasm. When p27^Kip1^ is sequestered to the cytosol, its nuclear activities are limited, with the opposite being true when localized to the nucleus. This shift in cellular location is primarily regulated by phosphorylation [65,66,67,68,69,70] by various kinases, which are discussed throughout the review.

AMPK-mediated phosphorylation of p27^Kip1^ on Thr170, Ser83 and Thr198 plays a role in cytoplasmic localization of p27^Kip1^ and promotion of cell survival [63]. Liang et al. reports AMPK-dependent p27^Kip1^ Thr198 phosphorylation promotes its sequestration to the cytosol, resulting in an increase in autophagy [5]. Under serum deprivation, it was found that both phosphorylation of p27^Kip1^ on Thr198 and p27^Kip1^ stability increased along with a concomitant induction of autophagy. It was determined that Thr198 phosphorylation and subsequent p27^Kip1^ stabilization is regulated by the AMPK pathway. In immortalized mouse embryo fibroblasts (MEFs), AMPK activation using AICAR resulted in a dose-dependent increase in p27^Kip1^ Thr198 phosphorylation. Additionally, an acute increase in p-Thr198 p27^Kip1^ was also seen in glucose-deprived 3T3L1 preadipocytes. Under both conditions, there was an increase in pAMPK, indicating that AMPK activation mediates p27^Kip1^ Thr198 phosphorylation and induction of autophagy [5].

The importance of AMPK-dependent p27^Kip1^ Thr198 phosphorylation on autophagy was further investigated using lentiviral mutants of p27^Kip1^ in muscle stem cells (MuSCs) [15]. Overexpression of the p27^Kip1(198A)^ mutant, which cannot be phosphorylated at the AMPK-specific Thr 198, resulted in increased cell death, even when treated with AICAR. Additionally, treating MuSCs with an AMPK inhibitor increased cell death; however, overexpression of p27^Kip1(198D)^, a Thr 198 phosphomimetic, rescued p27^Kip1^ and increased cell survival, further demonstrating the importance of AMPK-mediated p27^Kip1^ phosphorylation on autophagy [15].

As discussed previously, AMPK-mediated phosphorylation of p27^Kip1^ on Thr198 mediates its localization to the cytoplasm [5]. During times of metabolic stress, cytoplasmic p27^Kip1^ has been found to be pro-autophagic, enhancing cell survival, as well as anti-apoptotic, providing protection from cell death. Nuclear p27^Kip1^, however, inhibits cell cycle progression and makes the cell susceptible to quiescence, apoptosis, and senescence. Both Hiromura et al. and Liang et al. found that depletion of p27^Kip1^ in serum-deprived cells promotes apoptosis as indicated by rapid cell shrinkage, membrane dysfunction, and loss of cellular integrity [5,31]. To further demonstrate the relationship of p27^Kip1^ in apoptosis and autophagy, activation of the AMPK pathway led to the formation of autophagic vacuoles in p27^Kip1+/+^ MEFs, whereas it induced apoptosis in p27^−/−^ cells. The findings were similar in glucose-starved cells, in which autophagy occurred in p27^+/+^ MEFs, but failed to be induced in p27^−/−^ cells as indicated by lack of autophagic vacuoles [5]. These data indicate that AMPK-dependent phosphorylation of p27^Kip1^ mediates the interplay between apoptosis and autophagy based on subcellular location.

AMPK activity has been found to be hyperactivated in *Tsc2*-null tumors and cells [71], implicating them in the investigation of AMPK-mediated phosphorylation of p27^Kip1^ and cellular location. A study by Short et al. used a rat model for *Tsc2* deficiency and *Tsc2*^−/−^ cells to demonstrate AMPK-dependent localization of p27^Kip1^ to the cytoplasm provides resistance to apoptosis [63]. *Tsc2*^+/+^ MEFs were found to have low levels of activated AMPK and nuclear localization of p27^Kip1^, whereas *Tsc2*^−/−^ MEFS had high levels of p-AMPK and sequestration of p27^Kip1^ to the cytoplasm. Under serum deprivation, *Tsc2*^+/+^ MEFS demonstrated an increase in apoptosis, while the percentage of nuclei with apoptotic features decreased in *Tsc2*^−/−^ MEFs [63]. While these findings linked cytoplasmic localization of p27^Kip1^ with resistance to apoptosis, Campos et al. utilized p27^Kip1^ mutants and *Tsc2*-null cells to provide evidence that cytosolic p27^Kip1^ promotes rapamycin-induced autophagy and cell survival through AMPK signaling [62]. Together, these data further support the idea of AMPK-dependent sequestration of p27^Kip1^ to the cytoplasm provides resistance to apoptosis as well as enhances the autophagic process during periods of metabolic stress.

Interestingly, p27^Kip1^ has a role in cytoskeletal remodeling, including actin polymerization and microtubule assembly [4,72]. The majority of this research is from neurons and neuronal outgrowth [4]. However, the association between cytoskeletal remodeling and cellular autophagy could be related. Autophagy is often induced from a stress response resulting in cellular remodeling in an energetic and anatomical, i.e., cytoarchitectural sense. Although this is speculative, it would be conceivable that these two processes linked together to facilitate p27^Kip1^ to promote autophagy through downstream signaling and cellular remodeling through cytoskeletal reorganization.

## 5. p27^Kip1^ Function in Muscle Tissue

The cellular localization pattern of p27^Kip1^ has also proven to be important in regulating apoptosis and autophagy in stem cells. Stem cells use autophagy to renew cellular components to maintain normal cellular function [73]. Our group identified that translocation of p27^Kip1^ from the nucleus to the cytoplasm occurs in young activating muscle stem cells (MuSCs) [15]. Directly after isolation from the sedentary/unchallenged muscle, it was determined that p27^Kip1^ was localized to the nucleus; however, after 48 h of activation in culture, cytoplasmic p27^Kip1^ was dominant in young MuSCs. p27^Kip1^ mutants were used to demonstrate the importance of AMPK signaling and p27^Kip1^ localization. Overexpression of the p27^Kip1(198A)^ mutant, which cannot be phosphorylated at the AMPK-specific Thr198, was localized mainly to the nuclei, whereas the p27^Kip1(198D)^ mutant, a Thr198 phosphomimetic, was sequestered to the cytoplasm. Additionally, overexpression of the p27^Kip1(198D)^ mutant in young MuSCs increased autophagy, represented by double the number of LC3B puncta and a 2-fold increase in the ratio of LC3BII:I. There was no effect on MuSC autophagy from overexpression of the p27^Kip1(198A)^ mutant [15].

In cardiac tissue, p27^Kip1^ was found to protect metabolically stressed cardiomyocytes from apoptosis by promoting autophagy [74]. In addition, p27^Kip1^ gain-of-function studies showed increased cardiomyocyte autophagy, inhibition of apoptosis, improved cardiac function, and reduced infarct size following myocardial infarction [74]. To confirm autophagy as the mechanism of action, treatment with chloroquine, a small molecule inhibitor of autophagy, was able to negate these effects [74]. In cardiomyocytes, p27^Kip1^ is a direct target of miR-221, a microRNA that impairs autophagy and cardiac remodeling through suppression of p27^Kip1^ and activation of mTOR [75,76]. Overexpression of p27^Kip1^ was able to rescue autophagy in hearts from mice with cardiac-specific overexpression of miR-221 [75]. Similar findings were observed for miR-222 which also inhibits p27^Kip1^-induced autophagy in heart tissue and leads to cardiomyocyte inflammation and apoptosis [77]. Other investigations have determined that the location of p27^Kip1^ within cardiac tissue plays an important role in its function. Interestingly, cytosolic p27^Kip1^ located in mitochondria regulated beneficial effects on cell survival and apoptosis [78]. Moreover, mitochondrial p27^Kip1^ was shown to be both necessary and sufficient for cardiac myofibroblast differentiation [78]. Together, these findings demonstrate the importance of cytoplasmic p27^Kip1^ for cell survival and autophagy across various cell types, including those within muscle tissue.

## 6. Cytoplasmic p27^Kip1^ Regulates Autophagy and Cell Survival through mTORC1

Although the mechanism(s) by which p27^Kip1^ induces autophagy and cell survival are still not understood, evidence points to mTORC1 as a possible focal point. The mechanistic target of rapamycin (mTORC1) is the central mediator of basic cellular processes including cell metabolism, growth, proliferation, and survival in response to such factors as nutrients, growth factors, cellular energy, and stress. mTORC1 consists of raptor (regulated associated protein of mTOR), mLST8, DEPTOR, PRAS40 and mTOR [79]. As a key regulator of cell metabolism, mTORC1 controls the process of autophagy by regulating autophagosome formation. During times of nutrient availability, mTORC1 inactivates ULK1 and other proteins involved in autophagosome formation, which in turns prevents autophagy [80,81,82]. However, when cells undergo nutrient deprivation, mTORC1 is inactivated and autophagy can occur. The mechanism for p27^Kip1^ regulation of autophagy has been recently suggested to work through mTORC1 inhibition via interaction with LAMTOR1 [83,84], a protein in the Ragulator complex (Figure 2). p18/LAMTOR1 regulates the mTOR pathway in lysosomes as it is required for mTORC1 activation. Zada et al. demonstrated a connection between p18/LAMTOR1 and p27^Kip1^ during periods of nutrient shortage. MEFs in which p18/LAMTOR1 was ablated (p18^−/−^) were less susceptible to cell death and showed increased phosphorylation of p27^Kip1^ compared to p18^+/+^ cells under starvation conditions [83]. These data suggest that when LAMTOR1 is ablated, an increase in p27^Kip1^ is able to promote autophagy and cell survival in nutrient-deprived cells.

Nowosad et al. further defined the role of p27^Kip1^ in modulating autophagy via the mTORC1 signaling pathway during periods of metabolic stress [84]. The study utilized amino acid-starved p27^−/−^ MEFs to investigate the effect of p27^Kip1^ on autophagy under starvation conditions. It was found that mTORC1 signaling was higher in p27 ^−/−^ cells with a concomitant decrease in lysosomal activity and autophagy as a result of cytoplasmic sequestration of TFEB, a transcription factor that controls expression of lysosome-related genes. Following prolonged amino acid deprivation, it was discovered that a fraction of p27^Kip1^ localizes to lysosomes. Here, it binds to LAMTOR1 and prevents Ragulator assembly which is necessary for mTORC1 activation. Inactivation of mTORC1 in turn promotes autophagy via nuclear translocation of TFEB and subsequent induction of lysosome function [84]. Although cytoplasmic p27^Kip1^ inhibits mTORC1 activity, mTORC1 can indirectly phosphorylate p27^Kip1^ on Thr157 through the serum and glucocorticoid-inducible kinase (SGK1), resulting in cytosolic localization of p27^Kip1^ and promotion of cell cycle progression [70]. This supports the idea that multiple kinases target p27^Kip1^ for nuclear export for various functions including cell survival, autophagy and proliferation.

## 7. p27^Kip1^ and Aging

Delineating the effects of age on cell cycle inhibitors such as p27^Kip1^ is difficult due to their respective role in cell cycle control. This is evident in the p27^Kip1^ KO mice, as they show increased body size, multiple organ hyperplasia, and tumorigenesis [85,86]. In contrast, transgenic mice overexpressing p27^Kip1^ have decreased cell proliferation across many tissues and an associated progeroid phenotype, including loss of muscle mass. Interestingly, this phenotype was observed independent of increased markers of DNA damage response or cellular senescence [87]. Gene expression across tissues does not seem to increase with age in the mouse [87]. However, this particular study did not investigate cellular location of p27^Kip1^. It would be interesting if the “default” location of excess p27^Kip1^ is nuclear, as opposed to cytoplasmic.

Currently, the best-known regulator of p27^Kip1^ localization is AMPK, which phosphorylates and sequesters p27^Kip1^ to the cytoplasm. Aging reduces AMPK sensitivity with a subsequent impairment of downstream targets that regulate cellular energy homeostasis [88]. In particular, studies demonstrate that the ability to undergo autophagy declines with aging [89,90], resulting in an increase in cellular senescence. This phenomenon has been observed in various cell types including fibroblasts [91] and MuSCs, also known as muscle satellite cells [17,92,93]. We have summarized the impact of AMPK dysfunction on p27^Kip1^ signaling and autophagy in Figure 2. Satellite cell loss and/or dysfunction is correlated with impaired muscle regeneration during aging [92,94,95,96]. Moreover, aged MuSCs become susceptible to apoptosis, which could contribute to the reduction in satellite cell number with age [93]. In addition to the decrease in the number of satellite cells with aging, the diminished capacity to activate satellite cells appears to play a key role in development of age-related muscle loss, also known as sarcopenia [97]. It has been demonstrated that autophagy is required to provide energy to satellite cells for activation [98]. Garcia-Prat et al. observed that autophagic flux is decreased in quiescent satellite cells from aged mice, resulting in entry into senescence and decreased cell function [99]. It was shown that satellite cells from young mice contained an abundance of mature autophagosomes, whereas there was a complete absence of autolysosomes in old cells. When satellite cells were treated with bafilomycin, an autophagy inhibitor, blockage of autophagic flux was higher for old satellite cells compared to young. It was discovered that rescuing autophagy could reverse stem cell senescence and restore myogenic capacity in aged satellite cells as indicated by treatment with rapamycin, a potent inducer of autophagy [99].

As discussed previously, the protein Bcl-2 plays a significant role in the regulation of apoptosis and autophagy. Apoptosis involves the activation of intracellular proteolytic caspases with subsequent DNA fragmentation and breakdown of the cell, eventually leading to cell death. Bcl-2 is considered anti-apoptotic as it inhibits caspase activation, allowing for cell survival under prolonged periods of cellular stress. Jejurikar et al. observed that aged satellite cells express 40% less Bcl-2 than young and adult cells [100]. This finding was consistent with increased levels of programmed cell death in aged muscle stem cells. Satellite cells from the old group had a significantly higher quantity of activated caspases and fragmented DNA compared to young muscle stem cells. When treated with the pro-apoptotic factors TNF-α and Actinomycin D, a greater percentage of aged satellite cells underwent apoptosis compared to those from young muscle as indicated by increased presence of small, condensed nuclei and cellular membrane blebbing. Together, these findings demonstrate that satellite cells from aged muscle are more susceptible to apoptosis than young cells, which may contribute to impaired muscle regeneration during aging [100].

Studies investigating the effects of aging on the dynamics between autophagy and apoptosis have revealed that p27^Kip1^ plays a significant role in this interchange. In particular, subcellular location comes into play as a major regulator of p27^Kip1′^s function in the context of aging. It has been demonstrated that ectopic expression of p27^Kip1^ in satellite cells reduces proliferative capacity [101] and that satellite cells isolated from old muscle proliferate significantly less than cells from young growing muscles [16]. Although mRNA expression of p27^Kip1^ is lower in aged satellite cells [95], there is higher accumulation of p27^Kip1^ protein in the nucleus [16], where its ability to promote cell survival is limited indicating that nuclear p27^Kip1^ may play a role in decreased satellite cell function in old muscles.

Our group recently provided evidence linking age-related dysfunctional AMPK/p27^Kip1^ signaling with the propensity to undergo apoptosis over autophagy [15]. An inverse relationship between autophagy and apoptosis was observed in aged MuSCs. As seen previously, there was a decrease in proliferative ability as well as autophagy in aged MuSCs as represented by less LC3B puncta formation compared to young cells after a time course of physiological aging in culture. Apoptotic markers, including cleaved poly(ADP-ribose) polymerase (PARP), were significantly upregulated in aged cells. Furthermore, when autophagy was inhibited using lentivirus, apoptosis in old MuSCs increased 2-fold. This apoptosis-induction effect was not observed in young MuSCs following inhibition of autophagy. Across age groups, AMPK/p27^Kip1^ phosphorylation was reduced in old and geriatric MuSCs. When AMPK activation was rescued with AICAR, apoptosis could be prevented in aged cells. After determining the significance of AMPK signaling in the aged MuSC, downstream regulation of p27^Kip1^ was studied using p27^Kip1^ mutants. Overexpression of p27^Kip1(198D)^, a Thr198 phosphomimetic, was able to rescue the induction of cell death upon inhibition of autophagy, whereas overexpression of the p27^Kip1(198A)^ mutant, which cannot be phosphorylated at the AMPK-specific Thr198, could not produce the same effect. Interestingly, after 48 h in culture, p27^Kip1^ was located primarily in the nuclei in geriatric MuSCs compared to young MuSCs, in which the protein had translocated purely to the cytoplasm. Activation of AMPK signaling with AICAR allowed for translocation of p27^Kip1^ to the cytosol in geriatric MuSCs. These data indicate that aging may be responsible for attenuated AMPK signaling through preferential nuclear localization of p27^Kip1^ and promotion of apoptosis and senescence over autophagy and cell survival, which is promoted through cytosolic translocation of p27^Kip1^.

## 8. Conclusions

Once thought to only function as a cyclin-dependent kinase inhibitor, p27^Kip1^ is now regarded as a multifaceted protein, with roles in cytoskeleton rearrangement and dynamics, cell migration and motility, and apoptosis and autophagy. Evidence has shown that the aforementioned functions are highly dependent on the subcellular location of p27^Kip1^, whether it is nuclear or cytoplasmic. In cancer biology, for instance, p27^Kip1^ is considered a tumor suppressor in the nucleus, where it acts as a CDK inhibitor and has the ability to induce cell cycle arrest and inhibit cell proliferation. When sequestered to the cytoplasm, however, p27^Kip1^ is thought of as pro-oncogenic and promotes tumorigenesis [102,103]. Thus, inhibiting p27^Kip1^ localization to the cytoplasm may serve as a popular pharmacological target to prevent tumor progression [64]. Conversely, in the context of metabolic stress, cytoplasmic p27^Kip1^ is favored over nuclear p27^Kip1^ due to its capacity to promote autophagy and cell survival while in the cytoplasm. It is known that the cellular location of p27^Kip1^ is not static; it has the capacity to shift from the nucleus to the cytoplasm. When this ability to translocate to becomes diminished, such as during aging, cells become susceptible to apoptosis or senescence during conditions of stress (metabolic or other) due to nuclear sequestration. While strong evidence suggests that this process is regulated by the AMPK pathway, the mechanism(s) is not fully understood. Additional studies are needed to fully understand how aging affects the AMPK/p27^Kip1^ pathway and to identify potential therapeutic targets to promote autophagy and cell survival during the aging process.

## Figures and Tables

**Figure 1 cells-10-01430-f001:**
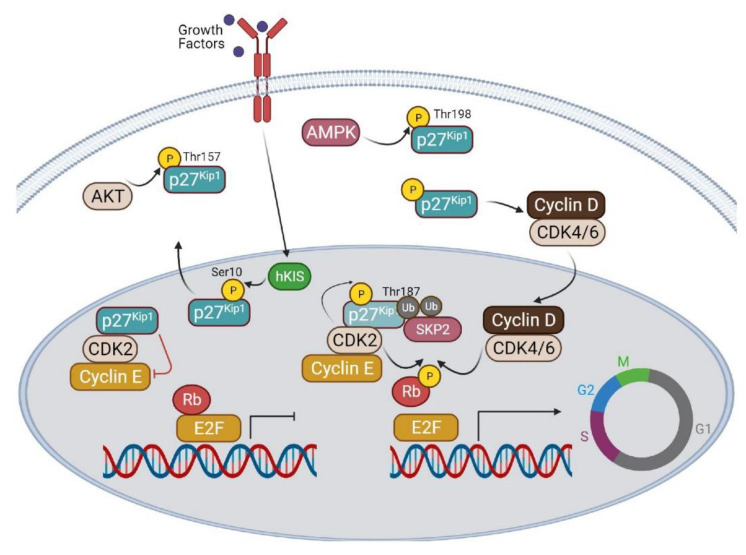
Cell cycle regulation of p27^Kip1^ and related signaling pathways. p27^Kip1^ can promote cell cycle arrest or proliferation, depending on the cellular location. Nuclear p27^Kip1^ will bind cyclin E and CDK2, inhibiting its kinase activity and leaving the *retinoblastoma protein* (Rb) protein unphosphorylated. In this state, the E2F transcription factor is inhibited from transcribing genes necessary to enter the cell cycle. With proper stimuli and/or growth factors, p27^Kip1^ is phosphorylated by CDK2 at Thr187, and is quickly ubiquitinated by SKP for subsequent proteasomal degradation. Without the inhibitor effects of p27^Kip1^, the cyclin E/CDK2 complex can phosphorylate, and inactivate Rb, allowing E2F to transcribe cell cycle genes. p27^Kip1^ can also be phosphorylated by hKIS at Ser10, which promotes sequestration to the cytosol. Cytosolic p27^Kip1^ can promote cell proliferation by importing cyclin D/CDK4/6 into the nuclei to further target Rb inhibition. Both Akt and AMPK will phosphorylate p27^Kip1^ on Thr157 and Thr198, respectively, to stabilize p27^Kip1^ in the cytosol and prevent reentry into the nucleus.

**Figure 2 cells-10-01430-f002:**
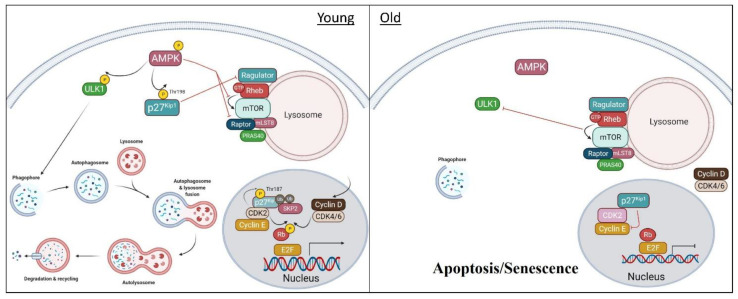
Age-related changes in p27^Kip1^ regulation and cellular function. In young cells, pro-survival and autophagy mechanisms are able to activate during cell stress and promote survival and proliferation. The balance between cell anabolism and catabolism is regulated, in part, by the AMPK/mTORC1 pathway. Under nutrient/energy stress, AMPK is phosphorylated, and activated from upstream signaling. This initiates a series of downstream signaling to ULK1 and p27^Kip1^ to promote autophagy and prevent apoptosis. AMPK-mediated phosphorylation of ULK1 with start the first step of the autophagy pathway, which is phagosome formation. Phosphorylated p27^Kip1^ will inhibit the pro apoptotic protein, Bax while inducing autophagy through inhibition of mTORC1. This is achieved by inhibition of ragulator, a key protein involved in docking mTORC1 regulatory proteins to the lysosome. In addition to the downstream effects, AMPK can directly inhibit mTORC1 by inhibition of mTOR kinase activity and through phosphorylation and inhibition of raptor protein. With proper AMPK activity, p27^Kip1^ will accumulate in the cytosol, stimulate autophagy and inhibit apoptosis. In addition, cytoplasmic p27^Kip1^ will recruit and import cyclin D/CDK4/6 into the nuclei so it can inhibit Rb and promote entry into the cell cycle. In aged cells, AMPK activation is suppressed during cellular stress. This results in hypophosphorylation of AMPK and loss of kinase activity. The inability to phosphorylate and stabilize cytosolic p27^Kip1^ will promote its sequestering in the nuclei and/or degradation of cytoplasmic p27^Kip1^. In the nuclei, p27^Kip1^ will inhibit cyclin E/CDK2 and suppress gene expression of cell cycle proteins. Furthermore, the lack of phosphorylated ULK1 and p27^Kip1^ leads to a reduction in autophagy and susceptibility to apoptosis. The lack of mTORC1 inhibition will further inhibit ULK1 and autophagy.

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
