# Peer review of "The AMPK/p27Kip1 Pathway as a Novel Target to Promote Autophagy and Resilience in Aged Cells"

_cells, 2021, doi:10.3390/cells10061430_

Round 1

Reviewer 1 Report

The present manuscript is well-structured, well-written and easy to understand.

This title is p27Kip1: A novel target to promote autophagy and rejuvenate aged cells.

As we all know, p27Kip1 is now known to function as a multifaceted protein, with roles in cytoskeleton rearrangement and dynamics, cell migration and motility, and apoptosis and autophagy.

Cytoplasmic p27Kip1 has been found to be pro-autophagic;

Nuclear p27Kip1 inhibits cell cycle progression and makes the cell susceptible to quiescence, apoptosis.

We consider p27Kip1 could switch the cell fate with the same function as Caspase-2.

Caspase-2 has been implicated in both apoptotic and non-apoptotic signaling pathways, including tumor suppression and cell cycle regulation; caspase-2 activation platforms; a cytoplasmic platform that is P53-induced protein with a death domain (PIDD) independent, and a nucleolar platform that requires PIDD, each providing access to distinct substrates that regulate cell fate by both pro-apoptotic and non-apoptotic mechanisms.

In the below two sections, the author didn’t well-addressed the questions.

Interplay between apoptosis and autophagy

p27Kip1 in apoptosis and autophagy

and how the Nuclear p27Kip1 and Cytoplasmic p27Kip1 translocation, this is a very important part, the author should list a section to discuss.

And many many references not cited. Such as:

FASEB J. 2019 Jan; 33(1): 1235–1247.

Stabilization of p27Kip1/CDKN1B by UBCH7/UBE2L3 catalyzed ubiquitinylation: a new paradigm in cell-cycle control

p27Kip1 Is Required to Mediate a G1 Cell Cycle Arrest Downstream of ATM following Genotoxic Stress.Erica K. Cassimere, Claire Mauvais, Catherine Denicourt. PLoS One. 2016; 11(9):

The totally reference number must be more than 120.

Author Response

Reviewer #1

The present manuscript is well-structured, well-written and easy to understand.

This title is p27Kip1: A novel target to promote autophagy and rejuvenate aged cells.

As we all know, p27Kip1 is now known to function as a multifaceted protein, with roles in cytoskeleton rearrangement and dynamics, cell migration and motility, and apoptosis and autophagy.

Cytoplasmic p27Kip1 has been found to be pro-autophagic;

Nuclear p27Kip1 inhibits cell cycle progression and makes the cell susceptible to quiescence, apoptosis.

We consider p27Kip1 could switch the cell fate with the same function as Caspase-2.

Caspase-2 has been implicated in both apoptotic and non-apoptotic signaling pathways, including tumor suppression and cell cycle regulation; caspase-2 activation platforms; a cytoplasmic platform that is P53-induced protein with a death domain (PIDD) independent, and a nucleolar platform that requires PIDD, each providing access to distinct substrates that regulate cell fate by both pro-apoptotic and non-apoptotic mechanisms.

In the below two sections, the author didn’t well-addressed the questions.

Interplay between apoptosis and autophagy

p27Kip1 in apoptosis and autophagy

and how the Nuclear p27Kip1 and Cytoplasmic p27Kip1 translocation, this is a very important part, the author should list a section to discuss.

Author’s Response: We have added to and re-structured these sections to better address the importance of p27 subcellular location as it relates to apoptosis and autophagy. We believe the revised draft is easier to follow in respect to their respective topics.

And many many references not cited. Such as:

 FASEB J. 2019 Jan; 33(1): 1235–1247.

Stabilization of p27Kip1/CDKN1B by UBCH7/UBE2L3 catalyzed ubiquitinylation: a new paradigm in cell-cycle control

p27Kip1 Is Required to Mediate a G1 Cell Cycle Arrest Downstream of ATM following Genotoxic Stress.Erica K. Cassimere, Claire Mauvais, Catherine Denicourt. PLoS One. 2016; 11(9):

The totally reference number must be more than 120.

Author’s Response: Our reference count is currently 107 and we believe we have thoroughly covered the literature as it pertains to our focused topic. After reviewing the references above we do not think they are relevant to this review and we have not included them in the current version.

Reviewer 2 Report

The manuscript entitled “p27Kip1: A novel target to promote autophagy and rejuvenate  aged cells” by Katz and White aims to review the knowledges on the role of p27Kip1 in the autophagic process and more specifically in the interplay between autophagy and apoptosis.

The chosen issue is certainly important and interesting. Indeed, p27Kip1 is a multifunctional protein, playing different roles in the different cellular compartments, some of them not CDK-related. Regarding autophagy, recently new pieces of information have emerged, making the review timely appropriate. The authors (White) also have published on the topic.

Comments

The authors strongly point to the cellular localization of p27Kip1 in determining the choice between autophagy and apoptosis. Particularly, AMPK appears as the key kinase induced in condition of serum starvation or nutrient (glucose) deprivation able to phosphorylate p27 in Threonine 198 keeping the protein in the cytoplasm. This causes stabilization and retention of p27 in the cytoplasm and activation of autophagy and inhibition of apoptosis. On the contrary, p27 activation of apoptosis may be exerted in the nucleus and is associated to its CDK-dependent activity.

Since the authors point to AMPK as central in the activation of autophagy in condition of stress and nutrient deprivation a more general description of its role and targets would be helpful;

Indeed, to make the manuscript clearer and more comprehensive, there are few points that I suggest to address further:

 - the role of AMPK in activating autophagy in condition of stress should be dissected in major detail;

- the mechanism by which the AMPK/ (pT198)p27 axis is able to stimulate autophagy is scarcely discussed and/or speculated;

- it has been reported that AMPK is able to phosphorylate p27 in other residues (S83, T170): they are not mentioned in the manuscript;

- other residues play role in the cytoplasmic relocalization of p27, such as S10 (phosphorylated by KIS, DIRK1, AKT) and T157 (phosphorylated also by AKT): they are not mentioned at all;

- T198 is also target of other enzymes, such as AKT, RSK1, playing roles in different processes. This should be mentioned and commented. It would also be interesting to discuss the intersection of the cytoplasmic proautofagic role with the other cytoplasmic p27 functions, such as cytoskeleton remodeling and vescicular trafficking;

-- the role (both in the nucleus and in the cytoplasm, and CDK-dependent and independent) of p27 in controlling apoptosis should be better described and discussed.

Other points of criticisms:

- Few times in the text and in the figure legends, the single letter amino acid code Y is used for indicating Threonine. Y is the code for Tyrosine. It has to be replaced by T.

Author Response

Reviewer #2

The authors strongly point to the cellular localization of p27Kip1 in determining the choice between autophagy and apoptosis. Particularly, AMPK appears as the key kinase induced in condition of serum starvation or nutrient (glucose) deprivation able to phosphorylate p27 in Threonine 198 keeping the protein in the cytoplasm. This causes stabilization and retention of p27 in the cytoplasm and activation of autophagy and inhibition of apoptosis. On the contrary, p27 activation of apoptosis may be exerted in the nucleus and is associated to its CDK-dependent activity.

Since the authors point to AMPK as central in the activation of autophagy in condition of stress and nutrient deprivation a more general description of its role and targets would be helpful;

Author Response: We have added some general background on AMPK under the AMPK and p27 heading, then lead into its interaction with p27

 Indeed, to make the manuscript clearer and more comprehensive, there are few points that I suggest to address further:

 - the role of AMPK in activating autophagy in condition of stress should be dissected in major detail;

Author Response: We discuss the AMPK signaling downstream to ULK and p27, two of its targets to promote autophagy, in paragraph 3 under AMPK and p27 heading

- the mechanism by which the AMPK/ (pT198)p27 axis is able to stimulate autophagy is scarcely discussed and/or speculated;

Author Response: Although this mechanism is still being understood, evidence points to mTORC1 as a focal point for p27 to induce autophagy. We have added a paragraph “Cytoplasmic p27Kip1 regulates autophagy and cell survival through mTORC1” where we highlight this pathway.

 - it has been reported that AMPK is able to phosphorylate p27 in other residues (S83, T170): they are not mentioned in the manuscript;

Author Response: These residues are mentioned in the 3rd paragraph under the AMPK and p27 heading. Since Ser83 and Thr170 residues have not been explored extensively, compared to Thr198, we don’t discuss their biology as much.

- other residues play role in the cytoplasmic relocalization of p27, such as S10 (phosphorylated by KIS, DIRK1, AKT) and T157 (phosphorylated also by AKT): they are not mentioned at all;

Author Response: We quickly discuss those residues in the introduction (lines 13-19).  However, the review focuses on AMPK signaling, so we focus our discussion through the respective targets.

- T198 is also target of other enzymes, such as AKT, RSK1, playing roles in different processes. This should be mentioned and commented. It would also be interesting to discuss the intersection of the cytoplasmic proautofagic role with the other cytoplasmic p27 functions, such as cytoskeleton remodeling and vescicular trafficking;

Author Response: The dual function of p27 is interesting and we thank the reviewer for bringing up this point. We have added a paragraph to the end of the “AMPK/p27Kip1 signaling promotes cell survival and autophagy” section. The paragraph reads:

Interestingly, p27Kip1 has a role in cytoskeletal remodeling, including actin polymerization and microtubule assembly [4, 76].  The majority of this research is from neurons and neuronal outgrowth [4], however the association between cytoskeletal remodeling and cellular autophagy could be related. Autophagy is often induced from a stress response resulting in cellular remodeling in an energetic and anatomical, i.e. cytoarchitectural sense.  Although this is speculative, it would be conceivable that these two processes linked together to facilitate p27Kip1 to promote autophagy through downstream signaling and cellular remodeling through cytoskeletal reorganization.

-- the role (both in the nucleus and in the cytoplasm, and CDK-dependent and independent) of p27 in controlling apoptosis should be better described and discussed.

 Author Response: We have described in more detail the role of p27 in apoptosis and added a section for its pro-apoptotic role in carcinogenesis:

Multicellular organisms undergo the physiological process of apoptosis, or programmed cell death, as a way to eliminate damaged cells and maintain cellular integrity. Apoptosis can result from cell cycle arrest in the late G1 or S phase [20-22]. p27Kip1 is known to function by blocking the transition from G1 to S phase and therefore plays an important role in apoptosis. When p27Kip1 was overexpressed in lung cancer cell lines, apoptosis was induced through downregulation of pRb expression [23]. Additionally, spontaneous apoptosis was significantly higher in p27-positive tumors from individuals with oral and oropharyngeal squamous cell carcinoma compared to p27-negative tumors. p27-positive tumors were also associated with higher levels of Bax expression, an apoptosis-related protein [24].  

Other points of criticisms:

- Few times in the text and in the figure legends, the single letter amino acid code Y is used for indicating Threonine. Y is the code for Tyrosine. It has to be replaced by T.

Author Response: We have replaced all single letter symbols with the correct 3 letter symbol in both text and figures.

Reviewer 3 Report

The manuscript by Katz and White provides an overview of the roles of p27Kip1 in the regulation of autophagy and apoptosis. The manuscript largely focuses on AMPK-mediated phosphorylation of p27 in this control and on work done in aging muscle stem cells.

It could therefore be appropriate to change the title to match more accurately the content of the manuscript by incorporating MAPK-p27Kip1 pathway and muscle or muscle stem cell in it.

The manuscript reads fairly well, the descriptions of some of the articles are a bit lengthy.

There are however a number of misleading statements that must be corrected.

-lane 32 “… its classical role in CDK-regulative activity…” This is awkward and should be changed.

-Lanes 40-41 “… while S10 phosphorylation by hKIS transports p27Kip1 out of the nucleus and contributes to actin organization and cortical neuronal migration via cytoplasmic stabilization of p27Kip1 [4, 14]. No, in Ref 4 the kinase phosphorylating p27 in neurons is CDK5. In ref 14, hKIS-mediated phosphorylation of p27 has been involved in cell cycle regulation, not migration or cytoskeletal organization. The authors should stick to what the articles actually show.

-Lane 67 “SKP” in the figure should be changed to SKP2 or the SCF-Skp2

-Lane 69 “p27Kip1 can also be phosphorylated by hKIS at S10 which promotes detachment of p27Kip1 from the CyclinE/CDK2 69 complex and translocation to the cytosol.” There is no evidence in Ref 14 (or elsewhere) that hKIS phosphorylation of p27 promotes the dissociation of p27 from Cyclin E-CDK2 complexes. In fact in Ref 14 it is stated that hKIS mediated phosphorylation of p27 takes place in G0-G1 before Cyclin E-CDK2 activation.

-Lane 84 “Bax functions by degrading p27Kip1 which…” There is no evidence in Ref 25 indicating that Bax degrades p27, this is misleading.

-Lane 86 “…suggesting that p27Kip1 reduces the activity of Bax” Same, no evidence in Ref 25 indicating that p27 regulates Bax activity

-Lane 91-92 “Autophagy is a process of degrading organelles and proteins to be used as substrates for additional cellular processes.” This is awkward, rephrase.

-Lane 113 “…and showed increased activation of p27Kip1” p27 has no “activity” something more specific must be used. Again on Lane 116-117.

-Lane 114 “Additionally, p18-/- cells had increased levels of autophagy-associated proteins, but these levels decreased when p27Kip1 expression was suppressed [35].” No, if the authors go back to ref 35 and look at the corresponding figure (4D), results show the opposite of what is said in the text of the article. In p18+/+ cells fail to induce autophagy (LC3BII) upon p27 siRNA, while p27 siRNA does not affect autophagy (LC3BII) in p18-/- cells.

-Lane 130 In this paragraph, the authors should mention/discuss the fact that AMPK also phosphorylates p27 on T170 and S83 and cite Ref 50 there.

-Lane 154 “…rescued p-Thr 198 p27Kip1…” Awkward as the authors are discussing the p27 T198D phosphomimetic mutant. Should be rephrased.

-Lane 174 “…p27Kip1 can prevent apoptosis by inhibiting the activity of Bax…” Again, there is no data in the literature showing this.

-Lane 177-178 “Mediation of cell fate during states of metabolic stress is an established function of p27Kip1 independent of its role as CDK inhibitory activity.” No, this regulation is not always CDK independent. In Ref 5 (Fig S4) there are data showing that the control of autophagy/survival in reponse to glucose starvation is at least in part CDK dependent. Same in Ref 25 where the effect of p27 on apoptosis in fully CDK dependent.

-Lane 184-185 “For instance mTORC can phosphorylate p27Kip1 on Thr157 through activation of the 184 serum and glucocorticoid-inducible kinase (SGK1),…” No, mTORC does not phosphorylates p27. mTOR activates SGK1 which phosphorylates p27. Rephrase.

-Lane 192, Ref 25 should also be cited there.

-Lane 195 “lead” correct to “led”

-Lanes 221-223: Authors should avoid using jargon like p27NT and actually say what the mutant is. In fact these mutants (p27NT= amino acids 1-93 and p27CT= 94-198) should be discussed keeping in mind the functional domains of p27. If p27NT is the one regulating autophagy in this article, this mutant only contains the cyclin-CDK binding domains of p27, while most “CDK-independent” roles of p27 have been attributed to the C-terminal part of p27.

-Lane229 “Our identified…” ???

-Lane 246, authors should cite and discuss PMID32123163 and PMID25394488

-Lane 254-255 This sentence makes no sense, missing a verb after “difficult”

-Lane 256 “contract” =contrast?

-Figure 2 the direct link between p27 and Bax on the left side should be removed, it is not direct.

-Lane 282 “phosphorylated” should be “phosphorylates”

-Lane 317 “Dynamic” should be “dynamics”

-Lane 360 “Thus, inhibiting p27Kip1 translocation to the cytoplasm serves as a popular pharmacological target to prevent tumor progression” No, this has not been demonstrated, should read as “may serve as a pharmacological target”

Author Response

Reviewer #3

The manuscript by Katz and White provides an overview of the roles of p27Kip1 in the regulation of autophagy and apoptosis. The manuscript largely focuses on AMPK-mediated phosphorylation of p27 in this control and on work done in aging muscle stem cells.

It could therefore be appropriate to change the title to match more accurately the content of the manuscript by incorporating MAPK-p27Kip1 pathway and muscle or muscle stem cell in it.

Author Response: Although we discuss recent work in muscle stem cells this review covers different cell types and is not intended to be specific to muscle. We do, however focus on the AMPK /p27 pathway. Therefor we will edit the title to:

The AMPK/p27Kip1 pathway as a novel target to promote autophagy and resilience in aged cells.

The manuscript reads fairly well, the descriptions of some of the articles are a bit lengthy.

There are however a number of misleading statements that must be corrected.

-lane 32 “… its classical role in CDK-regulative activity…” This is awkward and should be changed.

Author Response: This sentence has been reworded to “…its role in CDK inhibition and cell cycle control.”

-Lanes 40-41 “… while S10 phosphorylation by hKIS transports p27Kip1 out of the nucleus and contributes to actin organization and cortical neuronal migration via cytoplasmic stabilization of p27Kip1 [4, 14]. No, in Ref 4 the kinase phosphorylating p27 in neurons is CDK5. In ref 14, hKIS-mediated phosphorylation of p27 has been involved in cell cycle regulation, not migration or cytoskeletal organization. The authors should stick to what the articles actually show.

Author Response: This has been corrected to accurately reflect what the references show: “while Ser10 phosphorylation by hKIS promotes transport of p27Kip1 out of the nucleus allowing for cell cycle progression [14]. Additionally, Ser10 phosphorylation by CDK5 contributes to actin organization and cortical neuronal migration via cytoplasmic stabilization of p27 [4].”

-Lane 67 “SKP” in the figure should be changed to SKP2 or the SCF-Skp2

Author Response: This has been changed, the figure now shows SKP2 interacting with p27

-Lane 69 in the figure “p27Kip1 can also be phosphorylated by hKIS at S10 which promotes detachment of p27Kip1 from the CyclinE/CDK2 69 complex and translocation to the cytosol.” There is no evidence in Ref 14 (or elsewhere) that hKIS phosphorylation of p27 promotes the dissociation of p27 from Cyclin E-CDK2 complexes. In fact in Ref 14 it is stated that hKIS mediated phosphorylation of p27 takes place in G0-G1 before Cyclin E-CDK2 activation.

Author Response: We have edited this sentence to remove the reference to CyclinE/CDK2 complex. The sentence, located in figure legend #2, now reads:

p27Kip1 can also be phosphorylated by hKIS at Ser10 which promotes translocation to the cytosol.

-Lane 84 “Bax functions by degrading p27Kip1 which…” There is no evidence in Ref 25 indicating that Bax degrades p27, this is misleading.

Author Response: This statement has been removed.

-Lane 86 “…suggesting that p27Kip1 reduces the activity of Bax” Same, no evidence in Ref 25 indicating that p27 regulates Bax activity

Author Response: This statement has been removed.

-Lane 91-92 “Autophagy is a process of degrading organelles and proteins to be used as substrates for additional cellular processes.” This is awkward, rephrase.

Author Response: This sentence(s) have been reworded to:

Such substrates are generated through degradation of organelles and proteins. Damaged organelles and misfolded proteins are engulfed by autophagosomes and delivered to lysosomes for degradation. The released products are then recycled as an alternate energy source and substrates for protein synthesis.

-Lane 113 “…and showed increased activation of p27Kip1” p27 has no “activity” something more specific must be used. Again on Lane 116-117.

Author Response: “Activation” has been changed to “phosphorylation.”

-Lane 114 “Additionally, p18-/- cells had increased levels of autophagy-associated proteins, but these levels decreased when p27Kip1 expression was suppressed [35].” No, if the authors go back to ref 35 and look at the corresponding figure (4D), results show the opposite of what is said in the text of the article. In p18+/+ cells fail to induce autophagy (LC3BII) upon p27 siRNA, while p27 siRNA does not affect autophagy (LC3BII) in p18-/- cells.

Author Response: This statement has been removed.

-Lane 130 In this paragraph, the authors should mention/discuss the fact that AMPK also phosphorylates p27 on T170 and S83 and cite Ref 50 there.

Author Response: Phosphorylation of p27 on T170 and S83 are mentioned in this paragraph: “AMPK-mediated phosphorylation of p27Kip1 on Thr170 plays a role in cytoplasmic localization of p27Kip1.  AMPK may also be able to directly phosphorylate p27Kip1 at S83 as well [44].’

-Lane 154 “…rescued p-Thr 198 p27Kip1…” Awkward as the authors are discussing the p27 T198D phosphomimetic mutant. Should be rephrased.

Author Response: “p-Thr198” should not have been in this sentence and was removed.

-Lane 174 “…p27Kip1 can prevent apoptosis by inhibiting the activity of Bax…” Again, there is no data in the literature showing this.

Author Response: This statement was removed.

-Lane 177-178 “Mediation of cell fate during states of metabolic stress is an established function of p27Kip1 independent of its role as CDK inhibitory activity.” No, this regulation is not always CDK independent. In Ref 5 (Fig S4) there are data showing that the control of autophagy/survival in response to glucose starvation is at least in part CDK dependent. Same in Ref 25 where the effect of p27 on apoptosis in fully CDK dependent.

Author Response: The phrase “independent of its role as CDK inhibitory activity” was removed from this sentence.

-Lane 184-185 “For instance mTORC can phosphorylate p27Kip1 on Thr157 through activation of the 184 serum and glucocorticoid-inducible kinase (SGK1),…” No, mTORC does not phosphorylates p27. mTOR activates SGK1 which phosphorylates p27. Rephrase.

Author Response: This sentence was rephrased to: “For instance, mTORC can activate the serum and glucocorticoid-inducible kinase (SGK1), which phosphorylates p27Kip1 on Thr157 resulting in cytosolic localization of p27Kip1 and promotion of cell cycle progression [54].”

-Lane 192, Ref 25 should also be cited there.

Author Response: Citation added.

-Lane 195 “lead” correct to “led”

Author Response: Corrected.

-Lanes 221-223: Authors should avoid using jargon like p27NT and actually say what the mutant is. In fact these mutants (p27NT= amino acids 1-93 and p27CT= 94-198) should be discussed keeping in mind the functional domains of p27. If p27NT is the one regulating autophagy in this article, this mutant only contains the cyclin-CDK binding domains of p27, while most “CDK-independent” roles of p27 have been attributed to the C-terminal part of p27.

Author Response: p27NT and p27CT were clarified: “Two p27Kip1 mutants were used; p27NT (amino-terminus of p27) which localizes to the cytoplasm and p27CT (carboxy-terminus of p27) which demonstrates primarily nuclear localization.”

-Lane229 “Our identified…” ???

Author Response: Corrected sentence to: “Our group identified…”

-Lane 246, authors should cite and discuss PMID32123163 and PMID25394488

Author Response: Both articles are discussed: In cardiomyocytes, p27Kip1 is a direct target of miR-221, a microRNA that impairs autophagy and cardiac remodeling through suppression of p27Kip1 and activation of mTOR [58, 59]. Overexpression of p27Kip1 was able to rescue autophagy in hearts from mice with cardiac-specific overexpression of miR-221 [58]. Similar findings were observed for miR-222 which also inhibits p27Kip1-induced autophagy in heart tissue and leads to cardiomyocyte inflammation and apoptosis [60].”

-Lane 254-255 This sentence makes no sense, missing a verb after “difficult”

Author Response: This sentence has been edited to read:

Delineating the effects of age on cell cycle inhibitors such as p27Kip1 is difficult due to their respective role in cell cycle control.

-Lane 256 “contract” =contrast?

Author Response: Corrected to “contrast.”

-Figure 2 the direct link between p27 and Bax on the left side should be removed, it is not direct.

Author Response: That part of the figure is now deleted

-Lane 282 “phosphorylated” should be “phosphorylates”

Author Response: Corrected to “phosphorylates.”

-Lane 317 “Dynamic” should be “dynamics”

Author Response: Corrected to “dynamics.”

-Lane 360 “Thus, inhibiting p27Kip1 translocation to the cytoplasm serves as a popular pharmacological target to prevent tumor progression” No, this has not been demonstrated, should read as “may serve as a pharmacological target”

Author Response: Changed “serves as” to “may serve as.”

Round 2

Reviewer 1 Report

All comments were  well-addressed  at current version.

Author Response

We thank the reviewer for their comments and time. 

Reviewer 2 Report

The manuscript has been improved in this present version.

One minor point is that phosphorylations of p27Kip1 at T157, T198 by different kinases usually sequester p27Kip1 in the cytoplasm. It has not been clearly reported that they drive translocation from the nucleus into the cytoplasm. I suggest to revise the whole manuscript considering this point.

Author Response

We have gone through the manuscript and changed wording in some sections to change "nuclear to cytoplasmic translocation" to "cytoplasmic stability" and "cytoplasmic sequestering". We did leave the text referring to nuclear to cytoplasm translocation relating to T198, as our group has shown this site to regulate nuclear and cytoplasmic localization of p27. (White et al. 2018)

Reviewer 3 Report

The authors have addressed my concerns.

Author Response

(The authors gave the same response as above.)
